# TrafficMOT: A Challenging Dataset for Multi-Object Tracking in Complex Traffic Scenarios

## ABSTRACT

Multi-object tracking in traffic videos is a crucial research area, offering immense potential for enhancing traffic monitoring accuracy and promoting road safety measures through the utilisation of advanced machine learning algorithms. However, existing datasets for multi-object tracking in traffic videos often feature limited instances or focus on single classes, which cannot well simulate the challenges encountered in complex traffic scenarios. To address this gap, we introduce `TrafficMOT`, an extensive dataset designed to encompass diverse traffic situations with complex scenarios. To validate the complexity and challenges presented by `TrafficMOT`, we conducted comprehensive empirical studies using three different settings: fully-supervised, semi-supervised, and a recent powerful zero-shot foundation model Tracking Anything Model (TAM). The experimental results highlight the inherent complexity of this dataset, emphasising its value to drive advancements in the field of traffic monitoring and multi-object tracking.

## KEYWORDS

Traffic Video, Multi-object Tracking, Foundation Model

## 1 INTRODUCTION

Multi-object tracking (MOT) aims to detect the trajectories of multiple targets simultaneously along video sequences. It is a fundamental but indispensable task in diverse realistic application scenarios and has recently achieved significant advances in visual surveillance [16, 30], autonomous driving [4, 15, 22], AI city [18, 19, 27], and sports analysis [6, 17, 26, 33]. Among these diverse applications, multi-object tracking for traffic stands out, as it could unlock the potential for Intelligent Traffic Systems (ITS), which could enhance transportation efficiency and road safety [19].

Large-scale benchmarks are crucial for most artificial intelligence-related applications. Such benchmarks provide shared datasets for model training and fair competition in evaluation, e.g., the accident analysis dataset [24] and vehicle re-identification dataset [14], thus prompting the ITS development. Importantly, the development of effective multi-object tracking algorithms heavily also relies on the availability of high-quality datasets [9, 24, 27]. Despite the availability of multiple traffic MOT datasets, there are still significant limitations that hinder the progress in traffic analysis. For example, many existing datasets lack the complexity and diversity that is

*ACM MM, 2024, Melbourne, Australia*
© 2024 Copyright held by the owner/author(s). Publication rights licensed to ACM.
ACM ISBN 978-x-xxxx-xxxx-x/YY/MM
https://doi.org/10.1145/nnnnnnn.nnnnnnn

Table 1: Comparison of characteristics between existing traffic multi-object tracking datasets and `TrafficMOT`.

| DATASET | Year | #Video | #Mins | #Class | #Object |
|---|---|---|---|---|---|
| UrbanTracker | 2014 | 5 | 18.59 | 3 | 5.4 |
| CADP | 2018 | 1,416 | 86.37 | 6 | 3.6 |
| CityFlow | 2022 | 40 | **195** | 1 | 8.2 |
| ⋆ `TrafficMOT` | 2023 | **2,102** | 105.1 | **10** | **22.8** |

crucial for accurately reflecting real-world traffic scenarios, such as occlusions, varying lighting conditions, diverse traffic patterns, and importantly dense scenes. Consequently, the need for a new dataset arises, one that investigates the challenges encountered in complex traffic environments.

**Current Datasets in Traffic Analysis and Their Limitations.** Two widely used datasets in the field of traffic analysis are the VERI-Wild dataset [14] by Lou *et al.* and the MOTS dataset [28] by Voigtlaender *et al.* The VERI-Wild dataset focuses on vehicle re-identification in urban scenes and consists of data from 174 cameras. While it provides a comprehensive view of vehicle tracking scenarios, it is not specifically designed for multi-object tracking in intricate traffic scenarios. Similarly, the MOTS dataset extends multi-object tracking by incorporating object segmentation, but it also lacks a specific focus on complex traffic scenarios. Moreover, both datasets are not specifically designed for multi-object tracking in the context of intricate traffic scenarios.

Figure 1 and Table 1 provide the comparisons between our `TrafficMOT` and other datasets in the field of traffic multi-object tracking. Note that the Urban Tracker [9] is introduced in 2014, focusing on multiple object tracking in urban mixed traffic. However, this dataset consists of only five collected video sequences, limiting its diversity and scalability. Later, Shah *et al.* [24] present a CADP dataset for traffic accident analysis using CCTV traffic cameras. While the dataset contain 1,416 video segments with full spatio-temporal annotations, it is specifically tailored to accident analysis and does not cover the broader challenges of multi-object tracking in traffic. Recently, CityFlow [27] is introduced for vehicle tracking and identification, providing a valuable dataset for research in the field. However, the number of classes is limited and it is unable to handle dense traffic scenarios. Most recently, improved versions of CityFlow are released in [18], where a series of AI city challenges are held for multi-camera people tracking, tracked vehicle retrieval by natural language, and traffic safety [19]. While the above datasets are specifically tailored for MOT in traffic scenes, significant limitations still persist, including limited class numbers and traffic density.

**Contributions.** In this paper, we present `TrafficMOT`, a novel benchmark dataset that introduces new challenges in the field of multi-object tracking in traffic analysis. Acquired from fixed

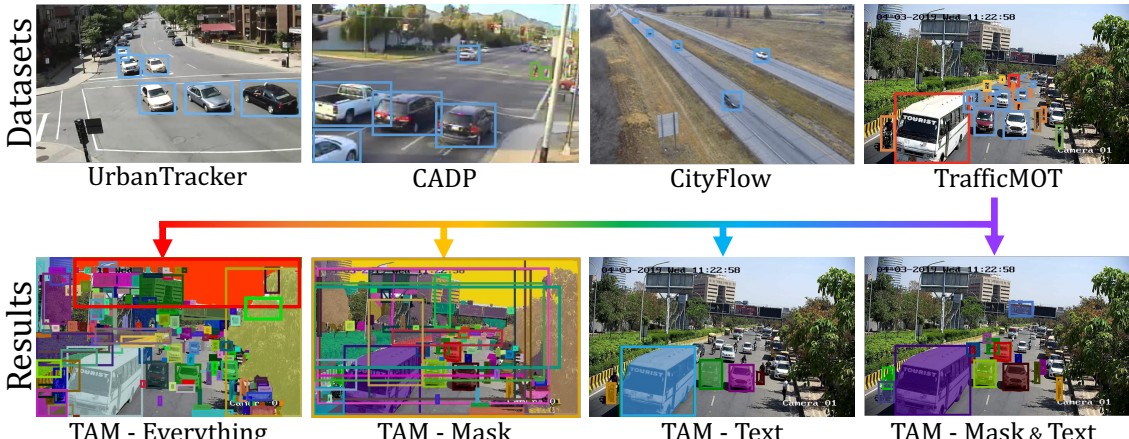

**Figure 1: Comparison of TrafficMOT with other traffic multi-object tracking datasets, and performance evaluations of the recent powerful zero-shot foundation model (Tracking Anything Model). The first row illustrates the enhanced complexity of our TrafficMOT, with more classes and instances. The second row presents the visual results in different settings of Tracking Anything Model. The visuals indicate that even the advanced model struggles to manage the complexities of our dataset.**

CCTV cameras across a wide range of cities, TrafficMOT stands out among existing datasets due to its unique and challenging characteristics. Notably, our dataset offers an extensive number of object classes, surpassing any other existing dataset. Additionally, TrafficMOT provides dense scenes, mainly capturing the intricate dynamics and congestion found in traffic scenarios. To highlight the complexity and challenges inherent in TrafficMOT, we also conduct an empirical study with three representative settings: fully supervised, semi-supervised, and a recent powerful zero-shot foundation model [10] named as Tracking Anything Model (TAM) [32]. Our experimental results underscore the inherent complexity of this dataset, demonstrating its potential to drive advancements in the field of traffic monitoring and multi-object tracking. Our contributions are summarised as follows.

- We introduce TrafficMOT, a novel benchmark dataset specifically designed for multi-object tracking in complex traffic scenarios. This dataset offers a larger number of object classes and encompasses intricate traffic situations, including densely populated scenarios.
- We conduct empirical studies with three representative settings: fully-supervised, semi-supervised, and a recent powerful zero-shot foundation model named as Tracking Anything Model (TAM). Experimental results highlight the intrinsic challenges posed by this dataset, thus emphasising its potential to spur advancements in traffic monitoring and road safety.

## 2 TRAFFICMOT: UNLOCKING MULTI-OBJECT TRACKING IN COMPLEX TRAFFIC

In this section, we delve into a comprehensive exploration of our dataset, providing detailed insights into its characteristics and statistical properties. We first introduce the data collection, annotation, and partitioning process. Then, we conduct statistical analyses to reveal the complexity of dataset.

### 2.1 TrafficMOT Benchmark Dataset Construction

**Video Collection.** We collected videos from eight cities spanning different regions in India, including northern, southern, eastern, western, and central parts of the country. The video acquisition process involved the use of fixed CCTV cameras installed on roads and highways. These cameras provided a comprehensive view of the traffic scenarios in different urban and suburban settings.

Our dataset consists of a total of 2,102 videos, where each video comprises 30 frames, allowing for a significant amount of temporal information to be extracted. To capture the heterogeneity of real-world surveillance systems, the videos were recorded using different cameras, resulting in varying frame resolutions, including 352 × 288, 1056 × 864, 1280 × 720, and 1920 × 1080. This diversity in frame resolutions reflects the practical challenges encountered in real-world traffic surveillance scenarios.

**Instance Annotation.** The annotation process for our proposed TrafficMOT dataset involved providing instance bounding box (bbox) labels and categorising each instance into one of the ten classes illustrated in Figure 2 for each annotated frame. Out of the total 2,102 videos in the dataset, 78 videos were fully annotated, meaning that all 30 frames in each of these videos were annotated. In these fully annotated videos, TrafficMOT provides tracking ids for each instance, enabling the tracking of objects across frames within the video. For the remaining 2,024 videos, annotations were only provided for the first frame. This resulted in a total of 8,689 unannotated frames across the dataset. The presence of this large number of unlabelled frames presents an opportunity to explore and advance semi-supervised learning techniques that leverage a limited number of annotations.

To ensure the quality of the annotations, a professional company was engaged to carry out the annotation process. Following the completion of the annotations, a dedicated quality control group

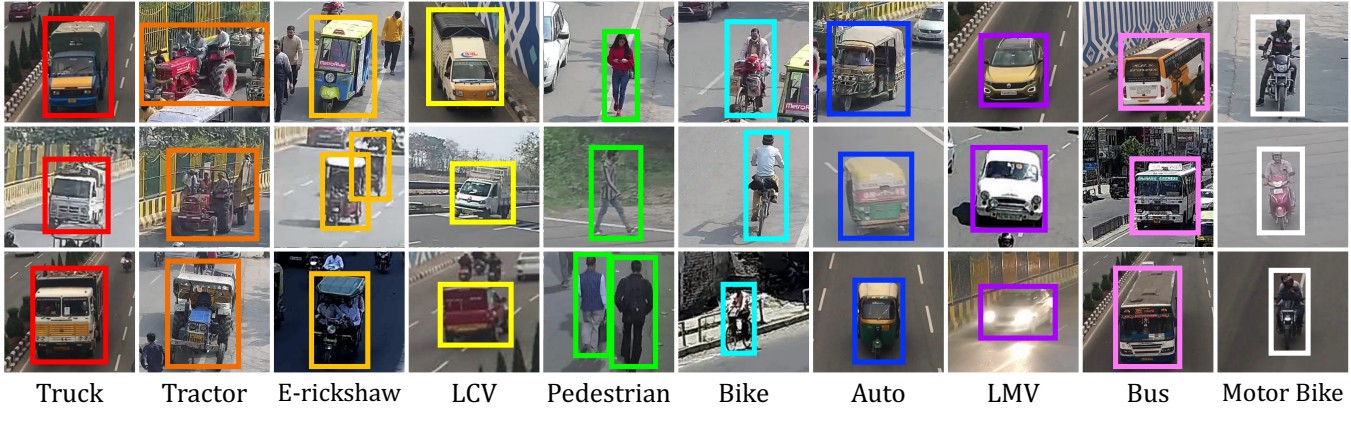

Truck · Tractor · E-rickshaw · LCV · Pedestrian · Bike · Auto · LMV · Bus · Motor Bike

Figure 2: Visualisation of annotated instances on different classes in TrafficMOT.

meticulously reviewed all the annotations. Any annotations identified as low quality were sent back to the company for refinement. This iterative process of annotation refinement was conducted multiple times to ensure a high level of annotation quality in the TrafficMOT dataset.

**Official Partition.** To facilitate the multiple object tracking task, the TrafficMOT dataset was divided into training and test sets at the video level. Out of the 78 fully annotated videos, 51 videos were selected as the testing set. These videos contain instance-level annotations (bounding box + category) for every frame and provide instance tracking IDs that persist throughout the video.

During the fully-supervised training process, we employed the remaining 27 fully annotated videos in conjunction with the 2,024 videos that only had annotations on the first frame. In contrast, the semi-supervised setting incorporated an additional 8,689 unlabelled frames as part of the training set. This inclusion of unlabelled data presents a chance to explore semi-supervised learning techniques that leverage both labelled and unlabelled data for improved performance. Partitioning the dataset into separate train and test sets ensures a rigorous evaluation of multi-object tracking performance and enables fair comparisons for different methods.

## 2.2 TrafficMOT Statistics & Key Features

**Diverse Classes.** Figure 2 presents annotation examples for each class, illustrating that certain classes exhibit a notable resemblance. For instance, distinguishing between the e-rickshaw and auto, as well as between the bike and motorbike, requires careful examination. Accurately identifying these classes is vital to ensure road safety, particularly considering the potential risks associated with the interaction between motor vehicles and non-motor vehicles. This underscores the importance of exploring complex traffic scenarios that involve interclass correlations.

In Figure 4, we present the correlation matrix depicting the presence of different classes within the same video. We divide the ten classes into two groups: motor and non-motor vehicles. The motor vehicle classes exhibit a strong correlation with each other indicating that different types of motor vehicles commonly co-occur in our dense and complex TrafficMOT dataset. Moreover, the correlation between motor vehicles and non-motor vehicles is

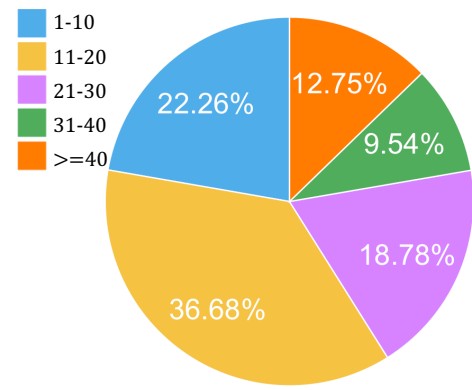

Legend: 1-10 · 11-20 · 21-30 · 31-40 · >=40

12.75% · 9.54% · 22.26% · 18.78% · 36.68%

Figure 3: The comparison of the number of instances per frame of our datasets.

shown in the pink region in Figure 4. It is evident that the pedestrian and e-rickshaws classes (0.37 and 0.47) frequently appear alongside heavy vehicles such as tractors raising safety concerns for citizens. Hence, our TrafficMOT dataset is beneficial for analysing dense traffic conditions and vehicle behaviour patterns to mitigate the risks of vehicle accidents.

**Crowded Scenes in Multi-Object Tracking.** Our dataset stands out from existing datasets in terms of traffic density, offering a unique opportunity to explore and tackle the challenges of multi-object tracking in dense traffic flow scenarios. While existing datasets are often limited in terms of the number of objects (as indicated in Table 1, and detailed in Figure 3 ), our dataset presents a significant advancement by providing a rich and challenging environment with densely populated traffic scenes.

A key characteristic of our dataset is the high average number of objects per frame. On average, our dataset contains 22.8 objects per frame, surpassing the ranges found in other datasets which average from 5.4 to 8.2 objects per frame, as shown in Table 1. This notable difference in object density creates complex tracking scenarios, requiring advanced algorithms and techniques to accurately track and analyse multiple objects amidst congested traffic conditions.

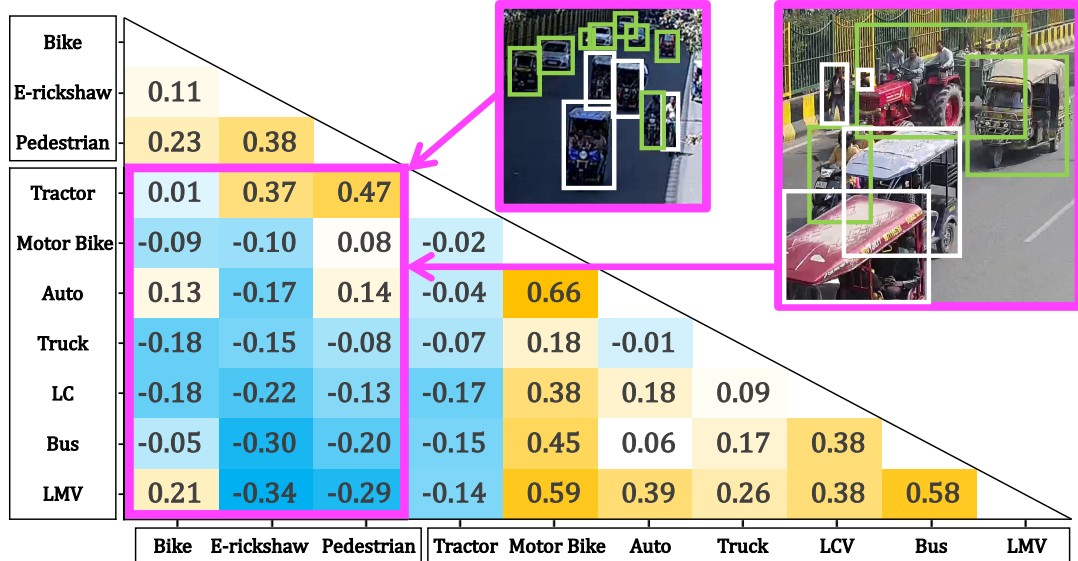

Figure 4: Correlation matrix of classes built on the appearance of class tracks within the same video.

*Why is Traffic Density Relevant?* The heightened object density in our dataset imposes greater demands on tracking algorithms, requiring them to exhibit enhanced robustness, accuracy, and efficiency in handling a larger number of objects simultaneously. Furthermore, the dense scene characteristic exposes tracking algorithms to challenges such as occlusions, partial object visibility, and overlapping trajectories, all of which are prevalent in crowded traffic scenarios. This realistic representation of traffic conditions provides a more meaningful and representative benchmark for evaluating the performance of multi-object tracking algorithms in practical traffic analysis applications.

**Weather Conditions in TrafficMOT.** Our dataset comprises videos captured under diverse weather conditions, providing a realistic representation of real-world scenarios. It encompasses sunny (daytime) weather (1,761 videos), foggy conditions (218 videos), and low-light conditions during nighttime (123 videos). This deliberate inclusion of various weather conditions enhances the training difficulty and exposes the model to a wider range of scenarios that it may encounter in the real world.

The presence of videos captured under sunny weather conditions reflects typical daytime scenarios, enabling the model to learn patterns and behaviours under optimal lighting conditions. This serves as a baseline for evaluating the model's performance in ideal weather conditions. Incorporating videos captured in foggy conditions introduces challenges associated with reduced visibility, as shown in Figure 5. The model learns to detect and track objects amidst the obscuring effects of fog, enhancing its ability to handle adverse weather conditions. The inclusion of videos recorded in low-light conditions during nighttime presents additional challenges, such as limited illumination and potential occlusions. This exposure allows the model to adapt and improve its performance in scenarios where lighting conditions are less favourable. By exposing the model to a diverse range of environmental conditions,

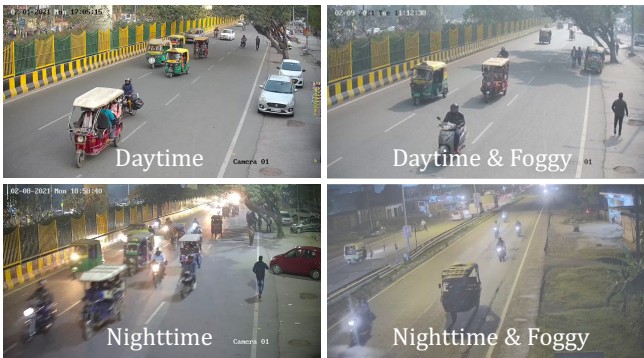

Figure 5: Visualisation of different weather scenes.

our dataset increases the training difficulty and enables the model to learn robust representations that generalise well across different scenarios. This prepares the model to handle real-world challenges, contributing to the development of more reliable and versatile multi-object tracking systems.

## 3 BENCHMARKING WITH TRAFFICMOT DATASET ON MULTIPLE OBJECT TRACKING

We conduct a comprehensive benchmark on multi-object tracking (MOT) by evaluating three types of methods: fully-supervised MOT methods (Section 3.1), semi-supervised MOT methods (Section 3.2), and a recent powerful zero-shot foundation model, the Tracking Anything Model (Section 3.3). This extensive evaluation allowed us to assess the performance of various MOT approaches on our `TrafficMOT` dataset and gain valuable insights into their respective strengths. It is worth noting that, to the best of our knowledge, *our*

*work is the first to apply large-scale foundation models to multi-object
tracking tasks in complex traffic scenarios.*

## 3.1 Fully-Supervised MOT Methods

We first train fully-supervised MOT models using existing baseline
MOT methods. Most baselines integrate an object detector followed
by a tracker, so the training process is formulated as

$$\mathcal{L} = \mathcal{L}_{det} + \mathcal{L}_{tra}, \tag{1}$$

where $\mathcal{L}_{det}$ represents the object detection and classification loss,
typically employed by methods like Faster R-CNN [23]. $\mathcal{L}_{tra}$ de-
notes object association or tracking loss, such as the loss used
for learning instance similarity [20]. In our study, we specifically
focus on four fully-supervised MOT methods: DeepSORT [29], Byte-
Track [34], QDTrack [20], and OC-SORT [2].

**DeepSORT** includes appearance information to SORT [1] to bet-
ter track occluded objects. SORT, which uses the overlap of bound-
ing boxes as a metric for object association, cannot effectively track
through occlusions due to the simple association metric. DeepSORT
thus integrates the appearance and motion information into the
association metric to further improve SORT.

**ByteTrack** makes full use of both the high- and low-score detec-
tion boxes for object association while most other MOT methods
only utilise the high-score ones. It first associates the high-score
detection boxes to certain tracklets based on their IoU similarity. Af-
ter that, it matches the low-score detection boxes to the remaining
unmatched tracklets.

**QDTrack** (Quasi-Dense Tracking) adopts a two-step approach
to enhance feature embedding and object association. Firstly, it
employs quasi-dense contrastive learning, which involves densely
matching extensive regions of interest in one image with corre-
sponding regions in another. This process aims to learn a more
robust feature embedding. Secondly, QDTrack uses bi-directional
softmax to associate objects, ensuring bi-directional consistency.
This means that the feature embeddings of two associated objects
should be the nearest neighbors of each other.

**OC-SORT** (Observation-Centric SORT) addresses noise accu-
mulation in SORT [1], which employs the Kalman filter (KF) and
an object association matrix. OC-SORT proposes two strategies:
Observation-centric Re-Update (ORU) and Observation-Centric Mo-
mentum (OCM). ORU rechecks untracked tracks associated with
object tracklets and updates KF parameters accordingly. OCM in-
troduces a cost matrix for object association, enforcing consistent
motion direction among objects.

## 3.2 Semi-Supervised MOT Training

In semi-supervised training setting, we incorporate the common
pseudo-labelling mechanism for object detection to implement
multi-object tracking. We apply three semi-supervised strategies
across all four MOT baseline methods, namely STAC [25], Soft-
Teacher [31], and MotionPrior [12].

**STAC** operates in two stages. In the first stage, an MOT model is
trained using labelled training data. In the second stage, the trained
MOT model is utilized to generate pseudo-labels for unlabelled data.
Subsequently, the MOT model is re-trained using both the labelled

and pseudo-labelled data, improving its performance through the
inclusion of additional unlabelled samples.

**SoftTeacher** shares a similar setting with STAC but introduces
a single-stage strategy to perform the two-stage pseudo-labelling
mechanism in an end-to-end manner.

**MotionPrior** extends STAC by incorporating simple motion
prior present in traffic datasets. Therefore, it can filter out noisy
pseudo labels, thereby enhancing the quality of the training dataset.

**Tracking Branches in Semi-Supervised Training.** The three
aforementioned pseudo-labelling strategies are primarily designed
for object detection, which may result in inaccurate pseudo-labels
for tracking IDs. Consequently, in the context of semi-supervised
MOT, it is necessary to ensure the consistency of pseudo-labels
across the time series before utilizing them for training purposes.

To this end, given a predicted object at a certain position specified
by bounding box $\hat{b}_i$, we will find the nearest object to $\hat{b}_i$ in the next
frame, which is denoted as $\hat{b}_{i+1}$. Then, we check the consistency $C$
of these two objects via

$$C = \begin{cases} 1, & \text{if } \text{IoU}(\hat{b}_i, \hat{b}_{i+1}) > \tau_c \text{ and } (\hat{y}_i == \hat{y}_{i+1}) \\ 0, & \text{otherwise} \end{cases} \tag{2}$$

where $\text{IoU}(\cdot, \cdot)$ is the function computing the intersection over the
union score of two inputs, and $\tau_c$ is a threshold. If the predicted
bounding boxes of these two objects are close to each other and
their object identities are the same, then $C = 1$. In this case, we
regard them as the same tracking object and keep their predictions
as pseudo-labels. In this way, we can obtain the pseudo-labels for
all the unlabelled videos, for both detection and tracking results.
Subsequently, we incorporate these pseudo-labelled videos along-
side the labelled ones to train another MOT model, employing the
same loss functions as defined in Equation (1).

## 3.3 Zero-Shot Foundation Model for MOT

Additionally, we conduct empirical studies using the Tracking Any-
thing Model (TAM) [32], a recent and powerful zero-shot founda-
tion model [10]. TAM demonstrates potential applicability in the
context of semi-supervised MOT, further enhancing the versatility
of our research.

**Track Anything Model**, also named Segment and Track Any-
thing (SAM-Track), is a powerful zero-shot foundation model for
general MOT tasks such as unsupervised MOT, semi-supervised
MOT, and interactive MOT. It adopts the latest Segment Anything
Model (SAM) [10] to generate a segmentation mask for the first
frame of videos with the help of user interactive clicks. Then it
exploits XMem [3], a semi-supervised MOT method requiring a
precise mask to initialise, to output a predicted mask. If the mask is
of poor quality, SAM is further used to refine the mask. Otherwise,
the mask is accepted as the final predicted tracking mask.

We consider four different ways to apply TAM for zero-shot
multi-object tracking, as shown in Figure 7. (a) We directly adopt
TAM to track everything on `TrafficMOT` using default settings. (b)
We make full use of the first-frame annotations in our dataset to
guide the learning process of TAM to track traffic-related objects.
(c) We input the class names in our dataset as the text prompts
to guide the TAM to learn to track the objects of these classes. (d)
We use both text prompts and first-frame annotations to better

Table 2: Numerical comparison on supervised settings. Best results are marked in bold.

| Techniques | | Evaluation Metrics | | | |
|---|---|---|---|---|---|
| Method | Year | mAP | mAP$_{50}$ | IDF1 | MOTA |
| DeepSORT | 2017 | 0.436 | 0.621 | 0.498 | 0.213 |
| ByteTrack | 2021 | **0.463** | **0.651** | **0.637** | 0.326 |
| QDTrack | 2021 | 0.440 | 0.633 | 0.592 | **0.414** |
| OC-SORT | 2022 | 0.460 | 0.646 | 0.562 | 0.118 |

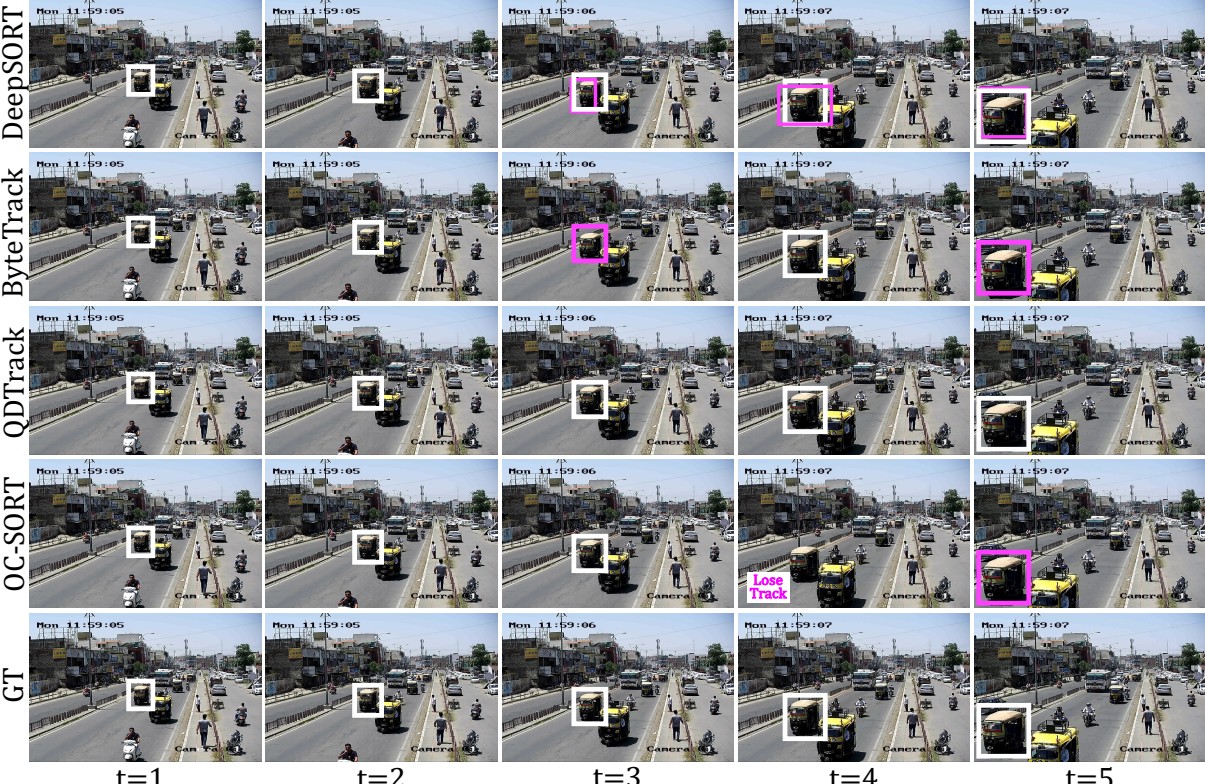

Figure 6: Visual comparison of different methods in fully-supervised Settings. For visualisation purposes, an object is displayed along with the identified error over time. The pink bounding box signifies a tracking error, indicated by a tracking ID that differs from the one assigned to the white bounding box.

adapt TAM to our dataset. Through these empirical studies, we illustrate that even a large-scale MOT model such as TAM struggles to effectively handle the MOT task as defined in our dataset. These comprehensive studies serve as a testament to the challenges posed by complex traffic scenarios within our dataset.

## 4 EXPERIMENTAL RESULTS

### 4.1 Implementation Details, Preprocessing, & Evaluation Metrics

**Implementation Details.** We implemented the MOT architecture based on MMTracking [5] framework using the PyTorch [21] library. For fair comparison with all the fully-supervised MOT methods [2, 20, 29, 34], we implemented the object detection module with the same architecture: Faster R-CNN (Region-based Convolutional Neural Network) [23] incorporating with region proposal network (RPN) [11] on ResNet-50 [8] backbone with ImageNet [7] pre-trained weights. The models were trained for 15 epochs with the AdamW optimiser [13] by step decay initialling with a learning rate of $2 \times 10^{-4}$. For the training of semi-supervised frameworks, we initialised their weights with the corresponding trained fully-supervised models. Further, we tuned the initial learning rate on AdamW optimiser to $5 \times 10^{-5}$ and trained an additional five epochs. The empirical studies on fully-supervised and semi-supervised

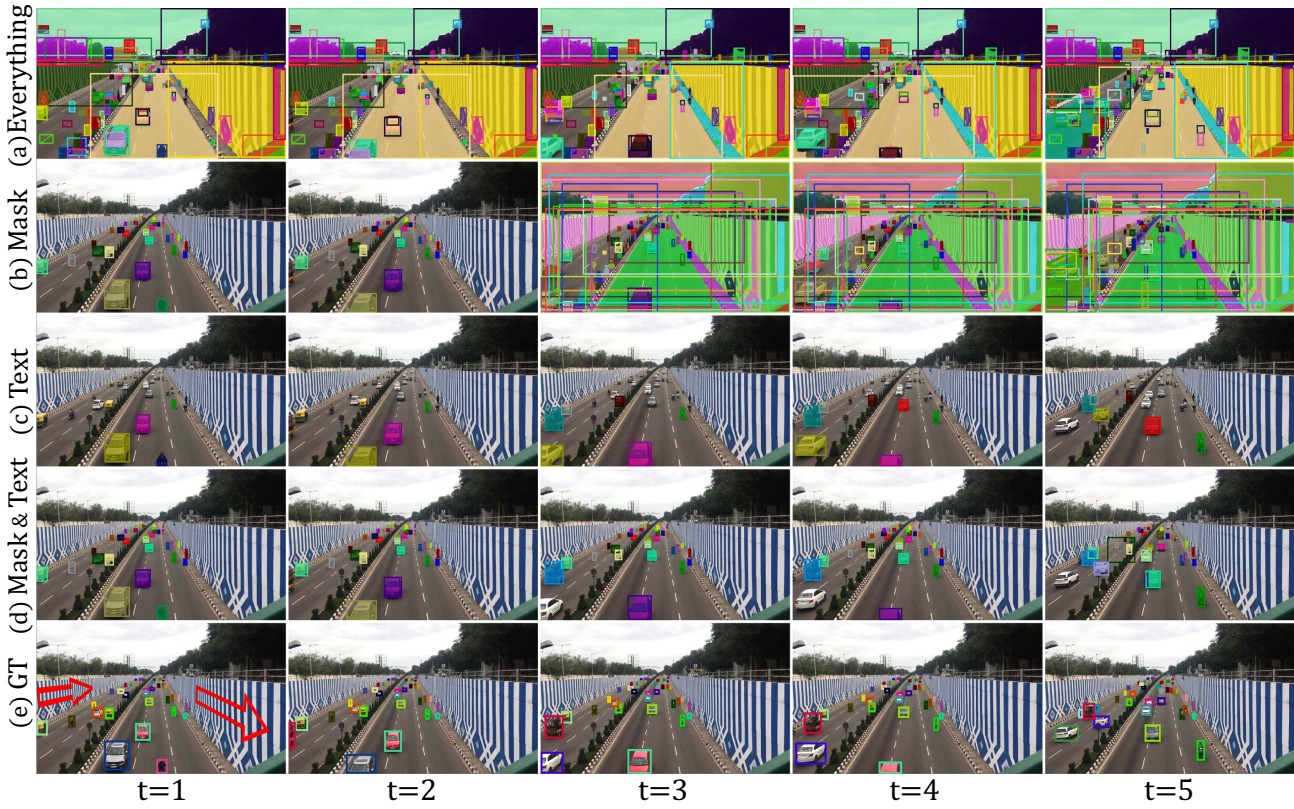

**Figure 7: Visual comparison using different information on TAM: (a) Everything, (b) Mask, (c) Text, (d) Mask & Text. For better comparisons of the outputs, the ground truth (GT) is displayed in (e). In the first frame of GT, we emphasise the two contrary road directions using red arrows.**

methods are all trained on an NVIDIA A100 GPU with 80GB RAM for approximately six hours and twlve hours respectively. For the TAM experiments, we set the new object updating parameters as default in all four track anything settings following [32].

**Preprocessing.** We followed the data augmentation strategy in MMTracking to enhance the variety of the dataset. Specifically, we first resized the images to $1,088 \times 1,088$ while preserving the aspect ratio, then performed photometric distortion, random flipping, normalisation, and padding.

**Detection Evaluation.** We measure the detection performance with the mAP (mean Average Precision) and $mAP_{50}$ metrics. mAP calculates the precision-recall curve by varying the detection threshold and averages the AP values for all classes, while $mAP_{50}$ takes the average precision at an IoU threshold of 0.5 for better evaluating moderate overlap objects.

**Tracking Evaluation.** We evaluate the tracking performance with Multi-Object Tracking Accuracy (MOTA) and Identity F1 score (IDF1). MOTA can assess the overall tracking accuracy by considering false positive, false negative, and tracking ID switches. IDF1 measures the accuracy of tracking identity by the F1 score based on the overlap between predicted and ground truth tracks.

## 4.2 Benchmark Results

**Fully-Supervised Results.** We benchmark four fully-supervised MOT methods, namely DeepSORT [29], ByteTrack [34], QDTrack [20], and OC-SORT [2]; as shown in Table 2 and Figure 6.

DeepSORT demonstrates relatively poor performance, mainly due to its challenges in handling occlusions, which are common complexities in traffic scenarios. On the other hand, ByteTrack [34] achieves the best performance in terms of mAP, $mAP_{50}$, and IDF1, with scores of 0.463, 0.651, and 0.637, respectively. In Figure 6, Byte-Track accurately detects bounding boxes but faces difficulties in establishing consistent connections across frames for tracking. Despite its limitations in detection, QDTrack outperforms ByteTrack in terms of MOTA by 0.088, indicating its superior ability to process tracking information. However, OC-SORT, being a newer method, exhibits below-average performance in terms of MOTA, suggesting it may not be well-suited for traffic scenarios. Figure 6 demonstrates instances where OC-SORT loses track. These results are obtained by re-training the networks with unified training parameters to ensure fair comparisons across different methods.

**Semi-Supervised Results.** In addition to benchmarking supervised methods, we also evaluate three semi-supervised pseudo-labelling strategies across all four baseline MOT models, the results of which are presented in Table 3.

**Table 3: Numerical comparison between baselines on semi-supervised Settings. The results display four metrics, with the best results highlighted in bold. "ST" stands for "SoftTeacher", and "MT" denotes "MotionPrior".**

**(a) Evaluation results on DeepSORT.**

| | RESULTS ON DEEPSORT | | | |
|---|---|---|---|---|
| | mAP | $\text{mAP}_{50}$ | IDF1 | MOTA |
| STAC | 0.378 | 0.565 | 0.504 | 0.333 |
| ST | 0.367 | 0.552 | 0.501 | 0.324 |
| MP | **0.390** | **0.569** | **0.511** | **0.340** |

**(b) Evaluation results on ByteTrack**

| | RESULTS ON BYTETRACK | | | |
|---|---|---|---|---|
| | mAP | $\text{mAP}_{50}$ | IDF1 | MOTA |
| STAC | 0.440 | 0.624 | **0.636** | 0.452 |
| ST | 0.436 | 0.601 | 0.601 | 0.450 |
| MP | **0.455** | **0.627** | 0.635 | **0.455** |

**(c) Evaluation results on QDTrack.**

| | RESULTS ON QDTRACK | | | |
|---|---|---|---|---|
| | mAP | $\text{mAP}_{50}$ | IDF1 | MOTA |
| STAC | 0.393 | 0.575 | 0.549 | 0.388 |
| ST | 0.393 | 0.569 | 0.530 | 0.346 |
| MP | **0.404** | **0.590** | **0.571** | **0.399** |

**(d) Evaluation results on OC-SORT.**

| | RESULTS OC-SORT | | | |
|---|---|---|---|---|
| | mAP | $\text{mAP}_{50}$ | IDF1 | MOTA |
| STAC | 0.422 | 0.610 | 0.603 | 0.385 |
| ST | 0.415 | 0.600 | 0.555 | 0.207 |
| MP | **0.444** | **0.629** | **0.610** | **0.397** |

Upon initial analysis, it is observed that MotionPrior generally outperforms the other two strategies, and STAC marginally surpasses SoftTeacher. When compared to the fully-supervised setting, all semi-supervised strategies yield lower scores in terms of mAP and $\text{mAP}_{50}$, which indicates reduced detection accuracy. However, models like DeepSORT, ByteTrack, and OC-SORT exhibit improved IDF1 and MOTA scores, which indicates improved tracking performance. Yet, even though the semi-supervised setting can help in improving tracking accuracy, the overall performance remains relatively low due to the dataset's complexity. Consequently, the design and development of further algorithms tailored to manage such challenging datasets is imperative.

**TAM Results.** Although TAM has demonstrated superior performance in interactive object tracking and segmentation in some videos [32], it yields unsatisfactory results on our `TrafficMOT` dataset, as depicted in Figure 7. We analyse the experimental results for five different TAM settings, as explained in Sec. 3.3. In the (a) track Everything mode, TAM outputs trajectories for all objects, including background and irrelevant objects, leading to a loss of focus on traffic-related objects. The second row in Figure 7 displays the tracking results when using the first frame annotation to guide the tracking of subsequent objects. It is evident that this model performs well only on the initial frames, and then transitions towards the (a) Everything mode for the remaining frames. From the third row, we observe that using class names as prompts allows TAM to generate stable tracking outputs for vehicles and pedestrians, but with numerous missing objects in crowded areas. When integrating the (d) first frame annotation and class name to guide TAM, the output outperforms all other settings. However, it may still generate unusual objects, as depicted by the large green bounding box in the last column. Moreover, new objects, such as the white car in the bottom-left corner visible in the third and fourth frames, are

not tracked until the fifth frame. A critical observation across all four settings is that while objects can be tracked, they are not accurately recognised. All instances are generalised as 'objects' without specific class designations. Despite TAM's promising pixel-level annotation, it falls short in providing precise class information.

## 5  CONCLUSION

This study addresses the vital area of multi-object tracking in traffic videos, recognising its potential to enhance traffic monitoring and promote road safety through advanced machine learning algorithms. We acknowledge the limitations of current datasets, which often oversimplify the task and lack adequate instances or diverse class representation. To tackle these challenges, we present `TrafficMOT`, a comprehensive and diverse dataset specifically designed to cover a wide range of complex traffic scenarios. Our research includes three empirical studies: fully-supervised, semi-supervised, and the recent zero-shot foundation model (TAM). These studies underscore the inherent complexity and difficulties of our `TrafficMOT` dataset. Our experimental findings highlight the inherent difficulty of our dataset while also showcasing its potential to drive significant advancements in the fields of traffic monitoring and multi-object tracking. With its high level of complexity and diversity, we envision that `TrafficMOT` will serve as a challenging yet realistic benchmark for the development and refinement of future multi-object tracking algorithms.

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
