# OpenReview forum: "TrafficMOT: A Challenging Dataset for Multi-Object Tracking in Complex Traffic Scenarios"
_acmmm.org/ACMMM/2024/Conference — MM2024 Poster_

### Official Review · Reviewer_h8w4 · 2024-05-24

**Rating:** 5
**Confidence:** 2

**Summary:**

TrafficMOT introduces a novel and challenging dataset that captures the intricacies of complex traffic scenarios, aiming to promote research and development in the field of multi-object tracking for traffic analysis and road safety applications. The dataset presents challenges such as object occlusions, diverse traffic patterns, and intricate interactions between motor and non-motor vehicles, reflecting real-world complexities in traffic monitoring. Comprehensive empirical studies are conducted using three settings: fully-supervised, semi-supervised, and a recent powerful zero-shot foundation model called Tracking Anything Model (TAM). The experimental results highlight the inherent challenges posed by the TrafficMOT dataset.

**Strengths:**

1) The introduction of the TrafficMOT dataset is a significant contribution. It addresses the limitations of existing datasets by providing a diverse set of videos from various regions, under different weather conditions, and with varying traffic densities. This makes it a valuable resource for advancing multi-object tracking (MOT) research in complex traffic scenarios. Besides, it includes real-world challenges such as occlusions, varying lighting conditions, and high object density. This realistic representation enhances the relevance and applicability of the dataset for practical traffic monitoring systems.

2) The paper provides a thorough description of the data collection, annotation process, and dataset partitioning. This level of detail ensures reproducibility and transparency, which are critical for benchmarking and further research.

3) The comprehensive empirical studies using fully-supervised, semi-supervised, and zero-shot foundation models (TAM) provide valuable insights into the performance of different MOT approaches on the TrafficMOT dataset. This helps in understanding the strengths and limitations of current methods.

**Limitations:**

1) The paper primarily focuses on the dataset and empirical evaluations but does not introduce novel algorithms or significant modifications to existing methods. This may limit its impact compared to papers that contribute new techniques or improvements.

2) The scale of the data is not significantly larger than before.

3) The high object density in the dataset poses scalability challenges for MOT algorithms. The paper could explore more scalable approaches or provide recommendations for handling such dense scenes more effectively.

4) The fully-supervised and semi-supervised methods used for evaluation are simple and outdated.

**Suitability:**

2

---

### Official Review · Reviewer_bWgp · 2024-05-24

**Rating:** 4
**Confidence:** 3

**Summary:**

This paper introduce TrafficMOT, a novel benchmark dataset specifically designed for multi-object tracking in complex traffic scenarios. This dataset offers a larger number of object classes and encompasses intricate traffic situations, including densely populated scenarios.

**Strengths:**

1. A multi-object tracking dataset in traffic scenes is proposed, and it is evaluated among multiple methods.
2. The evaluation of this dataset is comprehensive.

**Limitations:**

1. There is no contribution to algorithm design apart from the dataset.
2. There is limited introduction to the data collection process, lacking specific information on the number of samples for each category.
3. In terms of writing, the introduction should focus more on background and motivation, preferably without placing Table 1 in the introduction section.

**Suitability:**

2

---

### Official Review · Reviewer_vDD9 · 2024-05-25

**Rating:** 4
**Confidence:** 3

**Summary:**

This paper proposes TrafficMOT, a dataset for multi-target tracking in traffic scenes. Compared with previous traffic scene multi-target tracking data sets, TrafficMOT covers a wider range of categories, more complex scenes, and can better simulate real scenes. Afterwards, the author conducted experimental verification in the form of fully supervised, semi-supervised and zero-shot.

**Strengths:**

(1)TrafficMOT includes a large number of object categories and contains complex traffic situations. Compared with previous related data sets, TrafficMOT is more challenging and can better model the real traffic environment.
(2) TrafficMOT uses three representative settings (fully supervised, semi-supervised and zero-shhot foundation model) for experimental verification. The experiment proved the effectiveness of the mothod.

**Limitations:**

In the multi-target tracking data set of traffic scenes, BDD100K also contains many categories. Compared with BDD100K, what are the advantages of TrafficMOT?

**Suitability:**

3

---

### Official Review · Reviewer_pT8E · 2024-06-03

**Rating:** 2
**Confidence:** 4

**Summary:**

This paper presents a new dataset for multi-object tracking in traffic videos. The objects are annotated. Three different approaches are employed to benchmark the dataset.

**Strengths:**

Compared with existing datasets, more objects and classes are included and annotated in the proposed one.
Detailed annotations are provided.
Different settings are explored in the benchmark.

**Limitations:**

Basically this is still a relatively small dataset of only 105 minutes' videos. Compared with existing datasets, there is no significant difference. The dataset also doesn't pose any new problems for multi-object tracking.
Three existing approaches are employed with no novel approach being proposed.

**Suitability:**

2

---

### Meta-Review · Area_Chair_CP2p · 2024-06-25

**Recommendation:** Accept (Poster)
**Confidence:** 5

**Metareview:**

The reviewers have come up with the following strengths and limitations of the paper

STRENGTH
- Comprehensive and Detailed Annotations
- Diverse and Challenging Dataset
- Good Evaluation Settings
- Thorough Description of Data Collection and Annotation
- Significant Contribution to MOT Research

LIMITATIONS
- Small dataset size
- Lack of Novelty: The dataset does not introduce any new problems for multi-object tracking. Three existing approaches are employed, with no novel approach being proposed. There is no contribution to algorithm design beyond the dataset itself.
- Comparison with Existing Datasets: The paper does not highlight significant improvements or unique features that differentiate TrafficMOT from existing datasets.
- Limited data collection information
- Writing and Structure issues
- Scalability Challenges: The high object density in the dataset poses scalability challenges for MOT algorithms
- Outdated Evaluation Methods

None of reviewers joined the rebuttal and changed their rating. However, the rebuttal did answer some questions raised by the reviewers well.